# Utility of Combining High-Sensitive Cardiac Troponin I and PESI Score for Risk Management in Patients with Pulmonary Embolism in the Emergency Department

**DOI:** 10.3390/medicina59020185

**Published:** 2023-01-17

**Authors:** Elisa Cennamo, Gabriele Valli, Engy Khaled Mohamed Riead, Silvia Casalboni, Ilaria Dafne Papasidero, Francesca De Marco, Anna Mariani, Paola Pepe, Giuseppe Santangelo, Marina Mastracchi, Paolo Fratini, Giacinta Pistilli, Pasquale Pignatelli, Maria Pia Ruggieri, Salvatore Di Somma

**Affiliations:** 1Postgraduate School of Emergency Medicine, Faculty of Medicine and Psychology, University of Rome La Sapienza, 00185 Rome, Italy; 2Emergency Department, San Giovanni Addolorata Hospital, 00184 Rome, Italy; 3Department of Internistic, Anesthesiologic and Cardiovascular Clinical Sciences, University of Rome La Sapienza, 00185 Rome, Italy; 4Great Health Science Italy, 00144 Rome, Italy; 5Department of Medical-Surgery Sciences and Translational Medicine, University of Rome La Sapienza, 00185 Rome, Italy

**Keywords:** pulmonary embolism, emergency department, risk assessment, final disposition, PESI score, high-sensitive cardiac Troponin I

## Abstract

Background and Objectives: Pulmonary embolism (PE) has a major burden of morbidity and mortality, consequently the need for a prompt risk stratification for these subjects is crucial. In order to evaluate the risk management and final disposition of patients with PE in the Emergency Department (ED), we conducted a study that was divided in two phases: Phase I retrospective study (RS), Phase II prospective study (PS). Materials and Methods: In Phase I, 291 patients were enrolled while in Phase II, 83 subjects were evaluated. In both study phases, the enrolled subjects were analyzed for final disposition in ED using PESI score, right ventricle (RV) imaging, and high-sensitive cardiac troponin I (hs-cTnI) data. The RS patients were divided into low risk and high risk according to the sPESI score, while PS patients were grouped in low, intermediate, and high risk classes according to PESI score. In both study phases, all the studied patients were further divided into negative (hs-cTnI−) or positive (hs-cTnI+) groups according to hs-cTnI levels within normal or above cutoff values, respectively. For all enrolled subjects, CT pulmonary angiography was analyzed to assess the RV/LV diameter and volume ratio as an indicator of RV involvement. Results: In both RS and PS phases, hs-cTnI+ group showed a higher PESI score. Nevertheless, a significant percentage of hs-cTnI+ patients resulted to be in the low-risk PESI class. Patients with a positive RV/LV ratio were more likely to have a hs-cTnI+ (p < 0.01), while among those with a negative ratio, 24 to 32% showed as hs-cTnI+. In the hs-cTnI+ group from both study phases, patients were more likely to be admitted in an ICU (RR 3.7, IC: 2.1–6.5). Conclusions: In conclusion, in patients with PE in the ED compared PESI score alone, the combination of hs-cTnI and PESI seems to be of greater utility in improving risk stratification and final disposition decision-making.

## 1. Introduction

Venous thromboembolism (VTE), the pathophysiological entity that includes deep venous thrombosis (DVT) and pulmonary embolism (PE), represents the third most frequent acute cardiovascular disease after acute coronary syndrome and stroke, with an annual incidence between 39 and 115 cases in 100,000 inhabitants [1]. Furthermore, PE is a major cause of morbidity and mortality worldwide. 

Clinical guidelines and practice have been both widely implemented in the past decades, focusing the attention on the importance of risk stratification of these patients, in order to better plan the course of treatment [2]. However, there is still controversy on how to optimize patient’s risk management and final disposition for these subjects in the Emergency Department (ED) [3,4]. 

Strategies for risk assessment of acute PE patients have been of major interest during the past decades [5,6,7,8,9]. According to ESC 2019 guidelines [2], initial risk stratification is based on clinical symptoms and signs of hemodynamic instability, which represents the strongest predictor of early death [10]. Further stratification is necessary in patients without hemodynamic instability, to evaluate PE severity, searching for the presence of right ventricle (RV) dysfunction, or other concomitant aggravating conditions that may lead to a poor outcome [2].

Several scores have been validated to detect and evaluate the risk of death for cardiovascular events in PE [11,12,13,14,15]. The pulmonary embolism severity index score (PESI) and its simplified version (sPESI) are the most widely used, but they need to be integrated with the evaluation of RV involvement [13]. The PESI score focuses on demographic variables, comorbidities, and clinical parameters such as blood pressure, altered mental status, and oxygen saturation [13,14]. This approach allows the immediate estimation of patient severity but it does not take into account the possible subclinical myocardial involvement which instead has been shown to be closely correlated with mortality [16,17]. It must be accompanied by imaging of the RV (echocardiography or CTPA), searching for dysfunction. [2] Instead, according to current guidelines [2], in the acute phase of PE a troponin test is facultative and indicated to distinguish between intermediate-low and intermediate-high risk patients; while on the contrary RV imaging on ultrasound or CT scan, in order to detect RV involvement, is highly recommended. Nevertheless, when troponin is elevated in PE patients, the discharge from ED is contraindicated.

For the emergency physician, once PE is confirmed, it is crucial to immediately assess patient’s risk stratification in order to decide whether they need to be admitted into an intensive care unit (ICU) or intermediate intensive care unit (IICU), in a non-intensive ward (cardiology, internal medicine) or whether they can be discharged at home [2,4].

It is not clear how troponin levels change according to PESI risk class and no conclusive data are available on the incidence of elevated levels of high-sensitive cardiac troponin I (hs-cTnI) in the very low/low PESI risk class [18]. Indeed, this could result in a more accurate selection of patients eligible for home discharge [19].

The hypothesis of the study was that a significant percentage of patients from the very-low to the intermediate PESI risk group could have an elevated level of hs-cTnI and viceversa: leading to subsequent different clinical decision-making in ED. 

In particular we focused on troponin I because a recent study [20] showed that troponin T would be more strictly predictive of mortality in general, also for non-cardiac causes, while troponin I would be more directly correlated to myocardial damage, in particular ischemic damage. 

The aim of this pilot study was to verify, in patients presenting to ED with PE, whether the combination of PESI score with hs-cTnI evaluation plus imaging could be of greater utility compared to PESI score alone, in order to optimize patient’s risk-assessment and to improve the final disposition. 

## 2. Materials and Methods

Retrospective study (RS), Phase (I): a first retrospective study was conducted collecting data from 291 patients from January 2017 to January 2019 in two centers: Sant’Andrea Hospital and San Giovanni Hospital, both in Rome.

Prospective study (PS), Phase (II): a second phase of the study, from March 2019 to March 2020, at the ED of Azienda Ospedaliera San Giovanni Addolorata in Rome, Italy, was performed. A total of 83 consecutive patients (32 male and 51 female), presenting with a diagnosis of PE were enrolled in this study phase. 

We focused on two types of care-settings, patients that were admitted to the IICU that were high-risk patients, assessed on the basis of PESI, the extent of pulmonary embolism and exchange impairment, who require continuous 24-hour but non-invasive monitoring and who show no signs of hemodynamic instability or compromise requiring invasive treatments and monitoring (i.e., IOT or invasive BP monitoring).

The inclusion criteria in both studies were an age above 18 years old, and the diagnosis of pulmonary embolism in the setting of the emergency room (ER).

The exclusion criteria was the presence of acute coronary syndrome.

### 2.1. Study Design

For the PS phase, we collected at ED arrival from all the patients: personal data (name, gender, age), triage code [21], anamnesis, vital parameters such as blood pressure (BP), respiratory rate (RR), heart rate (HR), peripheral oxygen saturation (SpO_2_), Glasgow coma scale (GCS), temperature, routine blood tests, high-sensitivity cardiac troponin I (hs-cTnI) and arterial blood gas analysis (ABG).

The diagnosis of PE was made according to ESC guidelines [2]. Wells [22] and Geneva [23] scores were both used to assess the clinical pretest probability risk of PE. In patients with low risk for PE, according to Geneva and Wells score, the pulmonary embolism rule-out (PERC) [24] criteria was used in order to avoid unnecessary further tests for PE diagnosis; if PERC was negative, the patient was excluded from the study. Intermediate and high risk patients at the preclinical test underwent further deep analysis using d-dimer dosage when appropriate. Every suspected case was confirmed by a computed tomography pulmonary angiography (CTPA) for definitive diagnosis. 

Once the diagnosis of PE was confirmed, the PESI score was performed in order to stratify the mortality risk of the patient.

The PESI score consists of 11 items (age, gender, cancer, chronic lung disease, heart failure, heart rate > 110/min, systolic blood pressure <100 mmHg, respiratory rate > 30 min, body temperature < 36 °C, disorientation, lethargy, stupor, coma, SpO_2_ < 90%) that results in five classes of risk, meaning mortality risk in 30 days: I very low, II low, III intermediate, IV high, and V very-high risk. 

For statistical analysis, we summarized the five PESI risk classes into three classes, as follows: low risk patients (PESI class I-II), intermediate risk patients (PESI III), and high risk patients (PESI IV-V). 

In all subjects, within the first 24 h from the admission, we evaluated blood samples for hsTnI (Abbott Architect immunoassay [25], normal range: 0–15.6 ng·L^−1^ for women, 0–34.2 ng·L^−1^ for men). On the basis of a cut off level of hs-cTnI, patients were divided into: (a) negative (hs-cTnI−) within normal gender range or (b) positive (hs-cTnI+), higher than gender normal range, groups.

The RV involvement was assessed using two CTPA criteria [2] that are quantitative parameters of RV dysfunction: abnormally increased RV/LV diameter ratio on transverse sections and four-chamber views (RV/LV diameter ratio > 1 was considered as positive) and abnormally increased RV/LV volume ratio on transverse sections and four-chamber views (RV/LV diameter ratio > 1 was considered as positive). 

They were then analyzed in the two different groups: hs-cTnI+ or hs-cTnI−.

The final ward of hospitalization was collected for each patient and divided into three classes: high-intensive care ward (such as ICU or cardiologic ICU), mid-intensive care ward (IICU), and low-intensive care ward (LICW). 

For the RS phase, similar data as for the PS phase, were collected and analyzed except for the PESI score, that was not calculable in all patients and instead its simplified version (sPESI) was used. 

### 2.2. Statistical Analysis

The levels of hs-cTnI were related afterwards to the patient risk class that was obtained with PESI score. 

In order to verify the risk class of the patients, we chose a composite variable that included clinical worsening, rate of admission in intensive and intermediate care units, rate of need of orotracheal intubation, or death before admission.

For continuous variables the mean and the standard error has been reported. The normality of the data distribution has been evaluated using the Shapiro–Wilk test. The non-linear variables have been grouped in categories and it has been reported their frequency of occurrence and the percentage of the total number that were observed. The value of statistical significance (*p*-value) has been fixed at *p* < 0.05.

We grouped the patients with respect to the positivity or negativity of the hs-cTnI values (hs-cTnI+ group versus hs-cTnI− group) in accordance with the upper normal limits normalized for age, sex, and usual basal level of the patients (Abbott normal value: Trop M< 34.2 ng/L—F > 15.6 ng/L).

A one-way ANOVA test has been utilized in order to verify the presence of statistically significant differences between the different groups and statistically significant values of alfa were set at *p* value < 0.05. The source of variability has been evaluated by Tukey’s pairwise test. 

The nonlinear variables have been grouped in categories and their frequency of occurrence and the percentage of the total number observed have been reported. Correlations between non-parametric variables were tested by a $\chi^{2}$ test. 

The whole statistical examination was performed using the statistical software SPSS [IBM SPSS statistics, Version 20.0, SPSS Inc., Chicago, IL, USA].

## 3. Results 

### 3.1. Phase I

Retrospective study, Phase I: from January 2017 to January 2019, we retrospectively analyzed 291 patients. The patient’s characteristics of the retrospective phase are summarized in Table 1.

According to sPESI class, 28% of the Phase I cohort resulted in the low risk class (sPESI < 1). A history of cancer was present in 21% of cases, cardiovascular disease in 17% of cases, and a history of COPD in 9% of cases. High levels of hs-cTnI were present in 49% of cases at ED admission and the mean hs-cTnI was 187 ± 45 ng/L. 

A total of five patients died during the study period, all with positive values of troponin and all within the ED.

A significant statistical difference was found between hs-cTnI+ and hs-cTnI− groups and the RV/LV diameter ratio. A total of 54% of patients in the hs-cTnI− group had a positive ratio versus 82% of the hs-cTnI+ group (*p* < 0.001). The mean RV diameter was significantly higher in the hs-cTnI+ group, together with significantly lower values of LV diameters, compared to the hs-cTnI− group (Table 2). As shown in Figure 1, there is a positive relationship between hs-cTnI groups and sPESI risk class. In the low sPESI risk class, 73% of patients were in the hs-cTnI− group and 27% in the hs-cTnI+ group, compared with 42% and 58%, respectively, for the high sPESI risk class (*p* < 0.001).

No significant relationship was found between the two hs-cTnI groups and the intensity level of care of the admission ward.

### 3.2. Phase II

Prospective study, Phase II: from March 2019 to March 2020, 83 consecutive patients were analyzed. Patients’ characteristics of the prospective phase are summarized in Table 1, while laboratory results are summarized in Table 3. There were no significant differences in patient characteristics between Phase 1 and Phase 2 of the study.

In these 83 studied subjects at ED admission, the mean values of hs-cTnI were 88.8 ± 20.4 ng·L^−1^, in 50.6% of cases the concentration was over the upper limit, and the PESI score was 102 ± 4. According to the PESI score, patients were distributed as follows: low risk (39%), intermediate risk (25%), and high risk (35%).

In the prospective cohort, a previous history of cancer was present in 28% of cases, while in 15% and 17% of cases there was a history of cardiac diseases and COPD, respectively. Hypotension at ED arrival (SBP< 100 mmHg) was present in 6% of the cases. 

Table 4 compares the main physiological parameters and laboratory tests between the patients with an increased value of hs-cTnI above the upper limit (hs-cTnI+) and those with an hs-cTnI within the normal range (hs-cTnI−). 

Compared to hs-cTnI− subjects, the hs-cTnI+ group showed a significantly higher values of PESI score (PESI score, hs-cTnI+ vs. hs-cTnI−: 123 ± 5 vs. 80 ± 4, 1w-ANOVA: F 39.3, *p* < 0.0002). Grouping patients according with PESI risk class, the mean values of hs-cTnI were: 26 ± 8 ng·L^−1^, 49 ± 18 ng·L^−1^, and 178 ± 48 ng·L^−1^ (1w-ANOVA: F 6.69, dF 2;82, *p* 0.002), respectively, for the low, intermediate, and high risk groups. 

Figure 2A shows the distribution of hs-cTnI+ and hs-cTnI− patients in the three different PESI classes. A total of 83% of the patients with a high PESI risk class resulted in the hs-cTnI+ group and so did the 22% of patients with a low PESI risk class ($\chi^{2}$ 24.6 df 2, *p* < 0.0005). 

Considering together the hs-cTnI+ patients, 18% were in the low PESI risk class and overall 38% of hs-cTnI+ patients resulted in the low or intermediate PESI risk groups. Low PESI risk class patients who were in the hs-cTnI+ group had a RR 4.4 (IC 2.1–9.5) to be admitted to an intermediate or high intensive care unit. 

At CTPA evaluation, the hs-cTnI+ group showed significantly higher values of ratio RV/LV (Tabel 4), and frequency of RV involvement, i.e., positive RV/LV diameter.

It should be noted that 32% of patients with a negative RV/LV diameter ratio were in the hs-cTnI+ group. 

Patients in the hs-cTnI+ group were more likely to be admitted in a higher intensity care ward (i.e., ICU or IICU) than hs-cTnI− patients (Figure 2B) with an RR of 3.7 (IC: 2.1–6.5) to be admitted in an ICU or IICU compared to the hs-cTnI− group. 

A total of three patients died in ED before hospital admission. They were 95, 60, and 74 years old. The causes of death were cardiogenic shock for two patients and refractory respiratory failure for the other one. All the three of them showed an increase of hs-cTnI over the upper limit at admission (hs-cTnI+ group). One of them was in the intermediate PESI risk group and two in the high PESI risk group

## 4. Discussion

This study was conducted in two phases. The first was retrospective and a second one that was prospective, in subjects with symptomatic, acute pulmonary embolism in ED, and focused on optimizing patient’s risk stratification and management. 

Our results have shown that in all patients from the two study phases, a positive relationship between the PESI or sPESI risk class and the levels of hs-cTnI was found. 

Nevertheless, the percentage of patients that were classified in the low PESI or sPESI risk class presented elevated hs-cTnI levels in >20% of cases, therefore, the use of PESI score alone seems to not be adequate in detecting the presence of myocardial injury and so in the risk assessment process. 

The PESI score is influenced by age and its calculation relies on 11 different variables, so it is difficult to use in clinical practice and it wasn’t able to detect the presence of myocardial injury also in precedent studies [14,16].

The presence of myocardial injury, detected through an increase of hs-cTnI levels, has a clinical role in high-risk patients in predicting short-term death and adverse outcome events [26]. It also seems to move the patients to a higher risk for adverse events and poor outcome even in the context of a very-low/low PESI risk class [27,28]. This concept was extensively shown in the meta-analysis made by Barco and colleagues [19], which outlined that PESI low-risk patients are at higher risk of early mortality if myocardial injury and/or RV dysfunction is present. 

As such, it seems crucial to use a score that is able to properly classify the patients with a higher risk and detect the presence of myocardial injury at the same time [18,27], a task which the PESI score alone is not able to provide. 

The presence of myocardial stress or injury can be detected by some biomarkers by natriuretic peptides (BNP and NT-proBNP) or using troponin I and T (particularly a high-sensitivity assays) or even a new marker of damage, heart-type fatty acid-binding protein (HFABP) [29]. As already described in the introduction, troponin I seems to be more related with cardiac injury. Interestingly, troponin I is encoded in a genetic locus that is closely related to that of kallikrein-kinin axis proteins, with a close association with vasoactive peptides such as BNP [30] or endothelin 1 and pro-adrenomedullin [31], biomarkers that it is reasonable to imagine are released in conditions of wall stress and acute myocardial damage as they can occur following a sudden increase in right pulmonary hypertension or in conditions of sudden hypoxia due to the alteration of gas exchanges.

Troponin assessment, per se, already has already been shown to be a strong prognostic factor [28] in PE patients, but some validated scores already include biomarkers of myocardial injury in their stratification process (e.g., FAST Score [32] and BOVA Score [11,33]), but they are still not widely used and there is not a clear advantage in using one score system over the others [7,9]. 

Furthermore, although several studies have evaluated the prognostic value of cardiac troponins (determined by conventional assays) in acute PE [16,17,18,27,28], it has been supposed that conventional troponin assays may be not enough sensitive in ruling out an adverse outcome when performed during the first hours after symptom onset, thus requiring repeated measurements and resulting in a prolonged stay of the patient in the ER [29,34].

In this preliminary study, the presence of hs-cTnI upper to the normal range at ED arrival seems to be more sensitive in detecting the presence of myocardial injury and be able to select a more complex patient which needs a higher intensity of care, as shown in Figure 1. Notably, from our results, 83% of the patients that were admitted into an ICU had an hs-cTnI increased level as well as the 74% of patients who required a IICU; while on the other side, only the 14% that were admitted to an ordinary clinical ward presented an increase of hs-cTnI. Our findings are consistent with previous trials and meta-analysis on the prognostic role of troponin [26]. As a consequence, from our preliminary results it seems that in patients in the low PESI risk class but with an elevated hs-cTnI, discharging at home from the ED should be discouraged.

In PE, RV dysfunction is primarily defined by RV pressure overload and it can be assessed using imaging tests, both echocardiography or CTPA.

According to clinical guidelines^2^, CTPA can be used to stratify the patient risk regarding some parameters of RV involvement which involves also: left ventricle (LV) volume and diameter, right atrium (RA) and left atrium (LA) volume, a study of the inferior vena cava (IVC), i.e., RV/LV diameter ratio, RV/LV volume ratio, RA/LA volume ratio, contrast reflux into the IVC; these imaging methods have the same value of echocardiographic parameters assessed for the same purpose, but they are not routinely used in ED. 

Moreover, imaging methods are recommended to be part of the first steps in stratifying patients’ risk, instead troponin is facultative in intermediate and low risk patients. 

In our pilot study, CTPA parameters of RV involvement correlated with hs-cTnI rise but almost one third of patients who did not show significant changes in CTPA had a significant increase in hs-cTnI. CTPA parameters focus on a single cause that can lead to clinical worsening of these patients, the hemodynamic impairment of RV. It is likely that in these patients, myocardial injury is not simply a consequence of the RV hemodynamic impairment but also linked to other mechanisms such as: hypoxia, catecholamines release, changes in coronary perfusion, stress protein release or other underlying stressors. Therefore, hs-cTnI may be a more reliable and direct marker of myocardial damage and it may represent an independent predictor.

The study has some limitations. First, it is a single-center trial and the sample of patients that were enrolled is too small to delineate definitive conclusions and needs to be enlarged in order to better understand the relationship between hs-cTnI and PESI score and possibly their combined role in the prognostic stratification process. Lastly, we did not use a direct 30-day follow-up to confirm the poorer prognosis of hs-cTnI+ patients, we used instead an indirect method that was given by the setting of care that was needed by the patients in order to assess the severity of the patients. Here, an intrinsic bias could exist, even if the decision of admission and the intensity of care needed was not formally driven by the levels of hs-cTnI.

## 5. Conclusions

In subjects with PE presenting to ED compared to PESI score alone, combining hs-cTnI circulating levels and PESI score seems to be of greater utility in improving patient’s risk stratification and consequent decision-making. In these subjects, hs-cTnI plus PESI score assessment seems to be particularly suitable for more accurate final disposition, i.e., home discharge or admission to a more intensive care unit, because it is easy to obtain, fast in response, and highly reproducible.

Further studies with a larger number of patients and longer follow-up are needed in order to establish the correct timing in detecting the amount of myocardial injury that is mirrored by hs-cTnI levels in parallel with RV dysfunction detected by echocardiography or CTPA, coupling together cardiomyocyte cell death detection with RV dysfunction.

## Figures and Tables

**Figure 1 medicina-59-00185-f001:**
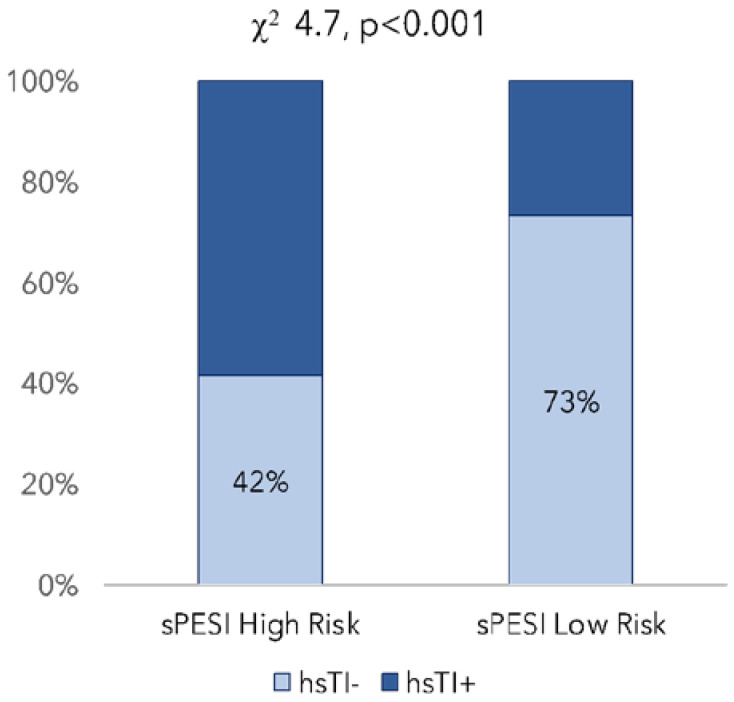
Comparison of the hs-cTnI+ group with the hs-cTnI− group in the sPESI risk class and sPESI low risk class respectively, in RS analysis.

**Figure 2 medicina-59-00185-f002:**
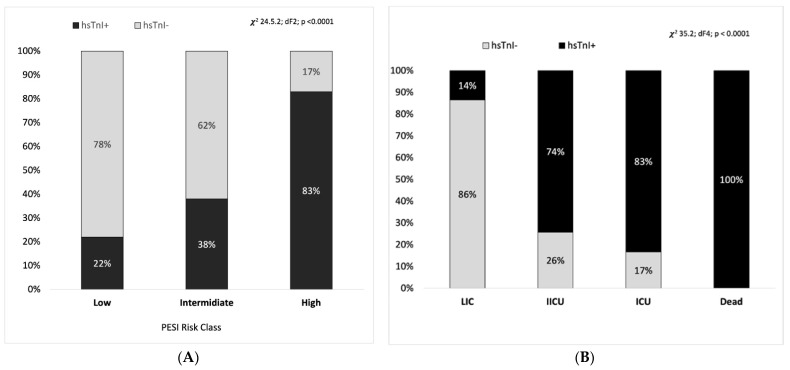
(**A**): Comparison of the hs-cTnI+ group with the hs-cTnI− group in the PESI low, intermediate and high risk class respectively, in PS analysis. (**B**): Comparison of the hs-cTnI+ group with the hs-cTnI− group in the different department of admission, grouped for the intensity of care (LIC: low intensive care; IICU: intermediate care unit; ICU: intensive care unit), in PS analysis; comparison of the hs-cTnI+ group with the hs-cTnI− group in patient died in ED, in PS analysis.

**Table 1 medicina-59-00185-t001:** Patients’ characteristics.

	Phase I	Phase II	P
n.	291	83	
Age, yrs (mean ± se)	70 ± 15	71 ± 16	n.s.
Gender, n (Female %)	149 (53%)	50 (61%)	n.s.
sPESI, % (high risk)	207 (71%)	54 (64%)	n.s.
SBP, mmHg (mean ± se)	129 ± 2	132 ± 3	n.s.
Hr, beat/min (mean ± se)	94 ± 1	92 ± 2	n.s.
RR, breath/min (mean + se)	21 ± 1	21 ± 1	n.s.
SpO_2_, % (mean ± se)	93.7 ± 0.3	94.1 ± 0.6	n.s.
MEWS, (mean ± se)	2.3 ± 0.3	2.9 ± 02	n.s.
RV diam, mm (mean ± se)	3.81 ± 0.04	3.85 ± 0.10	n.s.
LV diam, mm (mean ± se)	3.45 ± 0.05	3.75 ± 0.10	0.01
Ratio RV/LV, % (mean ± se)	1.16 ± 0.03	1.06 ± 0.03	n.s.
hs-cTnI, ng/L (mean ± se)	187 ± 45	89 ± 20	n.s.
ICU, n (%)	60 (21%)	11 (14%)	n.s.
Death, n (%)	5 (2%)	3 (3%)	n.s.

sPESI: simplified version of pulmonary embolism severity index; MEWS: modified early warning score for clinical deterioration; RV diam: right ventricle diameter; LV diam: left ventricle diameter; Ratio RV/LV: ratio right ventricle/left ventricle diameters; hs-cTnI: high sensitive cardiac troponin I.

**Table 2 medicina-59-00185-t002:** Comparison of the hs-cTnI+ group with the hs-cTnI− group in RS analysis.

	HS-TnI−	HS-TnI+	1w-ANOVA
Number of patients	148	143	-
Age, yrs ±SD	69 ± 1	70 ± 1	n.s.
MEWS	1.7 ± 0.3	3.0 ± 0.5	0.02
RV diam, mm	3.59 ± 0.06	4.03 ± 0.07	<0.001
LV diam, mm	3.68 ± 0.05	3.22 ± 0.07	<0.001
Ratio RV/LV	0.99 ± 0.02	1.34 ± 0.04	<0.001
Ratio < 1, %	46%	18%	<0.001
ICU, n (%)	17%	25%	n.s.
Death, n (%)	-	4%	0.02

MEWS: modified early warning score for clinical deterioration; RV diam: right ventricle diameter; LV diam: left ventricle diameter; Ratio RV/LV: ratio right ventricle/left ventricle diameters; ICU: intensive care unit.

**Table 3 medicina-59-00185-t003:** Laboratory results.

	Phase 1	Phase 2	*p* Value
Hb, gr/dL (mean + se)	13.4 ± 0.020	12.8 ± 0.2	n.s.
Creatinine, g/dL (mean + se)	0.986 ± 0.001	0.96 ± 0.04	n.s.
Hs-cTnI, ng/L (mean + se)	187.4 ± 0.481	89 ± 20	n.s.
D-dimer, μg/mL (mean + se)	5.4 ± 0.018	5.7 ± 0.8	n.s.
P/F ratio (mean + se)	n.a.	319 ± 10	n.s.
pH (mean + se)	n.a.	7.458 ± 0.007	n.s.
WELLS (mean + se)	3.7 ± 0.018	3.5 ± 0.3	n.s.
GENEVE (mean + se)	6.4 ± 0.012	7.4 ± 0.4	n.s.

EWS: modified early warning score for clinical deterioration; GCS: Glasgow coma scale; HR: heart rate; RR: respiratory rate; SBP: systolic blood pressure; Hb: hemoglobin; Hs-cTnI: high sensitive cardiac troponin I; P/F ratio: PaO_2_/FiO_2_.

**Table 4 medicina-59-00185-t004:** Patient’s parameters and laboratory test differences between the two groups: with a value of troponin within a normal range (hs-cTnI−) and those with a troponin level above the upper range (hs-cTnI+) in the prospective phase.

	hs-cTnI−	hs-cTnI+	*p* Value
Age, yrs ±SD	61.1 ± 16.3	76.9 ± 13.8	0.01
MEWS	1.50 ± 0.22	4.05 ± 0.29	0.0000001
SBP, mmHg	138 ± 4	126 ± 4	0.04
HR, beat/min	88 ± 2	96 ± 4	ns
RR, breath/min	19 ± 1	24 ± 1	0.00001
D-dimer, μg/mL	3.7 ± 0,8	7.7 ± 1,3	0.008
BNP, μg/mL	81 ± 23	437 ± 116	0.002
RV diam, mm	3.75 ± 0.17	3.92 ± 0.12	n.s.
LV diam, mm	3.48 ± 0.14	3.90 ± 0.12	<0.05
Ratio RV/LV	126 ± 0.05	0.99 ± 0.04	<0.001
Ratio < 1, %	18%	46%	<0.001
ICU, n (%)	15%	16%	n.s.

MEWS: modified early warning score for clinical deterioration; HR: heart rate; RR: respiratory rate; SBP: systolic blood pressure; BNP: brain natriuretic peptide.

## Data Availability

Not applicable.

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
