# Peer review of "Utility of Combining High-Sensitive Cardiac Troponin I and PESI Score for Risk Management in Patients with Pulmonary Embolism in the Emergency Department"

_medicina, 2023, doi:10.3390/medicina59020185_

Round 1
Reviewer 1 Report
1. Do you have clinical outcome data for both the retrospective and prospective groups?
2. Did the presence of raised Troponin in low-risk - intermediate risk predict poor outcome (i.e. subsequent ICU admission or death)?
- was there a difference in outcome in ICU/IICU in patient with raised vs normal troponin
3. Do you have data regarding
- upgrade of hospitalisation status (i.e. ward --> ICU)
- duration of ICU stay
- requirement for thrombolysis
4. Do you have similar data on BNP and does the combination of BNP and troponin help with risk assessment?
5. It is recognised that troponin is a marker of poor outcome. Low risk PE protocols worldwide utilise the troponin as a screening tool to exclude higher risk patients. However, the accuracy of troponin is not clear on its own. It would be useful to evaluate how troponin contributes to "down stepping" patients in ICU, especially in times where ICU capacity is strained
Minor:
1. Table 4. BPN should be BNP
Reviewer 2 Report
The study by Cennamo et al. investigated the importance of combining high-sensitive cardiac Troponin I and PESI score for risk management in patients with pulmonary embolism in the emergency department. In the retrospective and prospective studies, the authors tended to find out the relationship between hs-cTnI I and PESI score and possibly their combined role in the prognostic stratification process. It is an important clinical study that contributes to our knowledge about the prognosis and treatment of pulmonary embolism.
Comments: In the Introduction part, please provide more information about the importance of cardiac troponin T value in patients with pulmonary embolism. Why are the levels of high-sensitive cardiac Troponin I elevated by PE? How is it scientifically explained?
2 Were there any differences in the treatment of the patients before PE (oral anticoagulants, ACC etc.)?
3 In the Laboratory results Table 3 the retrospective group is missing, why?
4 How can you explain 2-fold difference in hs-cTnI value between retrospective (187 ng/l) and prospective group (89 ng/l) (Table 3)?
5 What are the main differences in the treatment strategy of PE between ICU and IICU? This information could be mentioned in the discussion.
